# Fatty Liver Disease-Alcoholic and Non-Alcoholic: Similar but Different

**DOI:** 10.3390/ijms232416226

**Published:** 2022-12-19

**Authors:** Stephen D. H. Malnick, Pavel Alin, Marina Somin, Manuela G. Neuman

**Affiliations:** 1Department of Internal Medicine, Kaplan Medical Center, Affiliated to Hebrew University, Rehovot 76100, Israel; 2In Vitro Drug Safety and Biotechnology, Department of Pharmacology and Toxicology, Temerity Faculty of Medicine, University of Toronto, Toronto, ON M5G OA3, Canada

**Keywords:** alcoholic steatohepatitis, nonalcoholic steatohepatitis, cytokines, microbiome

## Abstract

In alcohol-induced liver disease (ALD) and in non-alcoholic fatty liver disease (NAFLD), there are abnormal accumulations of fat in the liver. This phenomenon may be related to excessive alcohol consumption, as well as the combination of alcohol consumption and medications. There is an evolution from simple steatosis to steatohepatitis, fibrosis and cirrhosis leading to hepatocellular carcinoma (HCC). Hepatic pathology is very similar regarding non-alcoholic fatty liver disease (NAFLD) and ALD. Initially, there is lipid accumulation in parenchyma and progression to lobular inflammation. The morphological changes in the liver mitochondria, perivenular and perisinusoidal fibrosis, and hepatocellular ballooning, apoptosis and necrosis and accumulation of fibrosis may lead to the development of cirrhosis and HCC. Medical history of ethanol consumption, laboratory markers of chronic ethanol intake, AST/ALT ratio on the one hand and features of the metabolic syndrome on the other hand, may help in estimating the contribution of alcohol intake and the metabolic syndrome, respectively, to liver steatosis.

## 1. Introduction

There is clinical and epidemiological evidence on alcohol-induced damage to the liver. Ethanol’s metabolite acetaldehyde can cause DNA damage. Ethanol and its metabolites block DNA synthesis and repair by disrupting DNA methylation [1]. The alcohol dehydrogenase (ADH), cytochrome P-450 2E1 (CYP2E1), and catalase metabolize ethanol to acetaldehyde; acetaldehyde dehydrogenase (ALDH) enzymes then metabolize acetaldehyde to acetate. The oxidation of ethanol to acetaldehyde via ADH in the liver is connected with the reduction of NAD to NADH. NADH inhibits xanthine dehydrogenase activity. As a result, there is a shift of purine oxidation to xanthine oxidase, leading to the generation of oxygen-free radicals [1]. NADH mediates microsomal oxidations, via transhydrogenation to NADPH. Ethanol can also be reduced by inducible microsomal ethanol-oxidizing system. This induction is associated with an increased oxidation of NADPH with resulting H_2_O_2_ generation. Acetaldehyde forms DNA adducts and lead to mutations that are blocking DNA synthesis and repair. Both ethanol and acetaldehyde can disrupt DNA methylation by inhibiting S-adenosyl-L-methionine (SAMe) synthesis and DNA methyltransferase (DNMT) activity, impairing metabolism. Cytochrome P4502E1 (CYP)2E1 activity produces reactive oxygen species (ROS) leading to lipid peroxidation and formation of DNA adducts. Ethanol can also induce inflammation [1].

Histological manifestation cannot distinguish between the alcoholic liver disease (ALD) and non-alcoholic liver disease (NAFLD/NASH). The liver biopsy shows the presence of steatosis fibrosis and Mallory bodies (MB) [1]. NAFLD was first described by Ludwig et al. [2] in 1980. Ethanol can also induce inflammation and oxidative stress leading to lipid peroxidation and further DNA damage. One-carbon metabolism and folate levels are also impaired by ethanol. Other known mechanisms are discussed. Further understanding of the carcinogenic properties of alcohol and its metabolites will inform future research, but there is already a need for comprehensive alcohol control and cancer prevention strategies to reduce the burden of cancer attributable to alcohol.

It is the purpose of this article to review the similarities and differences between ALD and NAFLD.

## 2. Epidemiology and Clinical Features of NAFLD and ALD

NAFLD is the hepatic manifestation of the metabolic syndrome. The metabolic syndrome consists of obesity (increased visceral adipose tissue), systemic hypertension, dyslipidemia and insulin resistance or overt diabetes. In addition to obesity, other conditions that are associated with NAFLD include polycystic ovary syndrome [3], hypothyroidism, obstructive sleep apnea, hypopituitarism and hypogonadism [4].

NAFLD has a worldwide prevalence ranging from 6 to 35% [5]. The prevalence is increasing concomitant to the obesity epidemic. The majority of patients are diagnosed in their fourth or fifth decades [6] and the gender distribution is variable. Some reports suggest it is more common in females [2,6,7,8,9,10,11] and others in males [12,13,14,15,16,17]. Changing in lifestyle and obesity translate into increasing NAFLD-related cases of cirrhosis and hepatocellular carcinoma and related mortality [15]. Prevention efforts should be aimed at reducing the incidence of NAFLD and slowing fibrosis progression among those already affected [16].

There are also ethnic differences. In the United States there is a higher prevalence of hepatic steatosis in Hispanic Americans compared with Caucasian or Black Americans [13]. In Hispanics, the prevalence is linked to a greater prevalence of obesity. In Black Americans, the lower prevalence of fatty liver persisted after controlling for both body mass index and insulin resistance. Several systematic reviews refer to differences in lifestyle and epidemiology of NAFLD in the world.

## 3. The Importance of Fatty Liver Disease

The clinical picture of fatty liver disease is similar in many respects between NAFLD and ALD. Both are major causes of chronic hepatitis and cirrhosis. It has been estimated that by 2030, the prevalence of end-stage liver disease will increase by 2 to 3 times in Western countries and several Asian countries [18]. The diagnosis of fatty liver is based on the results of abdominal ultrasonography by trained technicians. All ultrasonographic images were stored in the image server. One gastroenterologist must review the images and make the diagnosis of fatty liver without reference to any of the participant’s other individual data. The four known criteria are hepatorenal echo contrast, liver brightness, deep attenuation, and vascular blurring.

The pathogenesis of NAFLD is linked to several risk factors. These include obesity, insulin resistance, hyperglycemia, diabetes, hypertension, dyslipidemia, genetics, aging, decreased physical activity and an aberrant intestinal microbiota. Obesity is by far the most important factor for the development of NAFLD. ALD also results in hepatic steatosis, steatohepatitis and fibrosis. In addition to alcohol intake, gender, genetics, the gut microbiota and other factors play a role. This will be discussed later in this review.

Alcohol use disorder is common worldwide with a prevalence in the United States of about 18%. Data from the National Health and Nutrition Examination Survey (NHANES), suggested the prevalence of alcohol-associated fatty liver disease amongst adults in the United States to be 4% [19]. The alcohol-associated liver disease mortality rate was estimated at 5.5 to 100,000 persons in 2012 [20] and 6.8 per 100,000 in 2018 [21]. The main risk factor is, of course, alcohol consumption above a certain threshold, which depends on age, gender and drinking pattern [22]. However, other factors that are linked to an increased risk for ALD include obesity [23], NAFLD, chronic viral hepatitis [24] and tobacco consumption [25]. 

## 4. Pathogenesis of ALD

The liver metabolizes the majority of ingested alcohol. It seems that the risk of ALD is linked to both the amount and duration of alcohol misuse [26]. Ethanol is initially converted to acetaldehyde via the enzyme alcohol dehydrogenase (ADH) [27]. 

Polymorphisms in ADH may be responsible for inter-individual variations in blood alcohol levels. Acetaldehyde is metabolized to acetate by mitochondrial acetaldehyde dehydrogenase (ALDH). Excess acetaldehyde impairs macrophage function, which impairs immune function [28]. CYP2E1-mediated metabolism leads to microsomal ethanol oxidizing system (MEOS) activation. MEOS induces highly reactive oxygen species (ROS) and hydroxyethyl radicals that contribute to enhanced oxidative stress.

In addition, there is ADH in the gastric mucosa; however, the activity is less in women than men [29]. This explains why women have higher levels of serum ethanol than men after similar consumption and, consequently, an increased risk for liver injury. Furthermore, *Helicobacter pylori* infection and gastritis can decrease the ADH activity [30].

The initial hepatic injury after exposure to alcohol is steatosis, resulting from reduced fatty acid oxidation and increased lipogenesis [31]. Chronic alcohol consumption upregulates lipogenic enzymes, including sterol regulatory element binding proteins (SRBEPs), and target genes including fatty acid synthetase acetyl CoA carboxylase and stearoyl-CoA desaturase [32]. In addition, the PLA3 (patatin-like phospholipase) gene variant influences hepatic fat accumulation [33].

ALD progresses from steatosis to steatohepatitis. This is a result of several factors including activation of innate immunity [34]. Alcohol-associated hepatitis (AH) results in an increase in neutrophils both peripheral and hepatic [35]. In addition, interleukins (IL)-8 and IL-33 are thought to have a role in neutrophils infiltration of the liver [35,36]. There is a role for macrophage migration inhibitory factor (MIF) in alcohol-associated liver disease but not in hepatitis C virus (HCV) infection [37].

The oxidation of ethanol results in the production of acetaldehyde and hydroxyethyl radicals, which covalently bind to proteins and results in the production of antigenic adducts. These adducts bind to crucial intracellular proteins resulting in cellular dysfunction [38]. Inflammatory cytokines such as IL-1 and IL-6 blood levels correlate with disease severity and liver histology [39,40]. The pro-inflammatory cytokine IL-17A is elevated in the AH [41].

Other mediators that have been implicated in the development of ALD include dietary fat [42] and oxidative stress [43,44,45]. The COX-2 enzyme produces prostaglandins [42]. The mitochondrial and cellular oxidative stress in chronic alcoholism makes hepatocytes susceptible to ethanol- or acetaldehyde-induced mitochondrial membrane permeability transition (MMPT). Necrosis and apoptosis are major cause of mitochondrial and cellular oxidative stress and damage in chronic alcoholism. Nitroso compound stress contributes to cell death by peroxynitrite formation [42]. The expression of the death receptor ligand CD95 is also upregulated by acetaldehyde metabolism [1,42,44]. Consequently, a dual mechanism, NADH-driven MMPT and CD95-mediated apoptosis, involving in both cases acetaldehyde metabolism and ROS production, are playing a role in ethanol-induced cell death. Mitochondria show increased H_2_O_2_ production and gluthatione (GSH) depletion and oxidation. Mitochondrial GSH plays a critical role in the maintenance of cell functions and viability and in mitochondrial physiology by metabolism of oxygen free radicals generated in the respiratory chain. GSH in mitochondria originates from cytosol by a transport system which translocates GSH into the matrix [1,43,45]. Dysfunctional hepatocytes present a loss mitochondrial cardiolipin and decreased mitochondrial membrane potential evolve to cell death. Mitochondrial oxidative stress precedes the initiation. 

MicroRNAs are noncoding RNA species that have important roles in many diseases. In alcoholic hepatitis, micro-RNA182 has been shown to be increased in cases with more severe disease, higher short-term mortality and hepatic ductular reaction [46]. In addition, miR-155 promotes steatosis in hepatocytes and is linked to increased TNF-α from macrophages [47,48]. There is a body of literature investigating the role of miRNAs in ALD and NAFLD. In addition, there is potential use of miRNAs as biomarkers and/or therapeutic targets. Among all miRNAs analyzed, miR-34a, miR-122 and miR-155 are most involved in the pathogenesis of NAFLD. Of note, these three miRNAs have also been implicated in ALD, reinforcing a common disease mechanism between these two entities and the effects of specific miRNAs. Currently, no single miRNA or panel of miRNAs has been identified for the detection of, or staging of, ALD or NAFLD [48]. Zhou team studied 416 genes in healthy controls and ALD patients. The miRNAs were screened. A miRNA-mRNA network was established; within this network, the miR-182-5p/FOXO1 axis was considered a significant pathway in ALD lipid metabolism. The miR-182-5p was found to promote hepatic lipid accumulation via targeting the FOXO1 signaling pathway, and inhibition of the miR-182-5p/FOXO1 axis improved hepatic triglyceride (TG) deposition in ALD by regulating downstream genes involved in lipid metabolism [49].

## 5. The Impact of Different Types of Alcohol on Alcohol Consumption

The quantity of alcohol and the types of alcoholic beverages impact human health by adding new converges. Preventative efforts should be aimed at reducing the incidence of the disease by decreasing alcohol consumption. Straka et al. [50] show the importance of the congeners in alcoholic beverages. They found an association between the drink, its specific taste, scent and color and these congeners. The specific taste of the drink has an influence on the consumer preference and consumption of alcohol. Alcohol congener’s interaction with the human organism follows different pathways in comparison to ethanol concentration.

It is likely that several drinks might contain additional substances enhancing the hepatotoxic effect of alcoholic beverages. The metanalysis presented data from seven cohort studies and two case-control studies (2,629,272 participants with 5505 cases of liver cirrhosis). There was no increased risk for liver cirrhosis in occasional drinkers, regardless the type of beverage [51]. In women only, consumption of one drink per day in comparison to long-term abstainers showed an increased risk for liver cirrhosis. Drinking ≥5 drinks per day was associated with a substantially increased risk in both women and men (relative risk [RR] = 12.44, 95% confidence interval [CI]: 6.65–23.27 for 5–6 drinks, RR = 24.58, 95% CI: 14.77–40.90 for ≥7 drinks) and men (RR = 3.80, 95% CI: 0.85–17.02, and RR = 6.93, 95% CI: 1.07–44.99, respectively). Heterogeneity across studies indicated an additional impact of other risk factors [51].

Detoxification, digestion, hormonal balance, blood sugar regulation and supporting the immune system are part of liver function. The disequilibrium of the functioning tasks is diagnosed with routine liver blood tests (ALT, AST, GGT) or tests required for treating other health conditions such as high blood pressure, high glucose and significant weight gain, type 2 diabetes mellitus, or stroke.

## 6. Pathogenesis of NAFLD

### 6.1. Insulin Resistance

The underlined cause of NAFLD is the metabolic syndrome [52]. NAFLD is regarded as the hepatic manifestation of the metabolic syndrome and the associated insulin resistance. Obesity and type 2 diabetes mellitus are both associated with peripheral insulin resistance and NAFLD. In addition, hepatic insulin resistance is often present in patients with lean NASH who have normal glucose tolerance [53]. However, not all patients with NASH have insulin resistance. There is a genetic influence on NASH.

A polymorphism in the gene encoding for apolipoprotein C3 [54] and IL-6 polymorphisms were associated with NAFLD, inflammation and insulin resistance [55]. Polymorphisms in PNPLA3, a phospholipase encoding for adiponutrin, a membrane bound protein, express primarily in adipose tissue exerting transacylase activity were associated with the histological severity of NAFLD, similarly to ALD [56]. Changes in the transcriptional activity of the peroxisome proliferator-activated receptor gamma co-activator 1-alpha (PPAR-gC1a) promoter has been found to be linked with the insulin resistance phenotype and NAFLD [57]. A single nucleotide polymorphism in PPARγ is linked to an increased risk for developing NAFLD in both children and adults [58,59]. 

An association between an increase in intrahepatic fat, increased gluconeogenesis, increased free fatty acid levels and insulin resistance [60] underlines the link between NAFLD and insulin resistance. Other cytokines and mediators associated with insulin resistance include IL-6 [61,62], TNF [63] and adipokines involved in insulin-receptor signaling [64]. As a result of insulin resistance, there is increased peripheral lipolysis, increased triglyceride synthesis and increased hepatic uptake of fatty acids [65]. As a consequence of this, there is a hepatic accumulation of triglycerides and, subsequently, beta-oxidation of free fatty acids (FFA) [66].

### 6.2. Hepatocellular Injury

Both in ASH and NASH, the injury is linked to the parenchyma. FFAs induce microsomal lipooxygenases generating hepatotoxic free oxygen radicals [67]. Patients with NASH, but not just hepatic steatosis, may have mitochondrial defects [68]. The mitochondrial function in NASH is influenced by under expression of genes important for mitochondrial function [69]. The development of hepatocellular injury requires the presence of both insulin resistance and managing free oxygen radicals. These include TNF-a, complement [70], plasma myeloperoxidase [71], natural killer (NK) cells [72], the hedgehog pathway [73] and caspase-2 [74].

The steatosis of the liver may be related to the metabolic syndrome or to excessive alcohol consumption, as well as the combination of one of these factors and medications.

Despite the different etiologies, the natural history of hepatic steatosis is similar. There is an evolution from simple steatosis to steatohepatitis, fibrosis and cirrhosis and its complications including hepatocellular carcinoma (HCC).

Liver pathology is very similar regarding NAFLD and ALD. In ALD, the liver presents lobular inflammation. There are characteristic morphological changes in the liver such as: organelle injury (mitochondrial loss of cristae), perivenular and perisinusoidal fibrosis, and hepatocellular ballooning [1].

### 6.3. Iron Overload

An increase in the concentration of iron in the hepatic parenchyma correlates with fibrosis severity [75,76]. Fletcher and Powell [77] consider that there is an added co-factor effect of iron and alcohol, both of which cause oxidative stress, hepatic stellate cell activation and hepatic fibrogenesis that triggers cirrhosis in these patients. Moreover, the cumulative effects of other forms of liver injury (viral or autoimmune) may result when iron and alcohol are present concurrently.

Furthermore, the pattern of iron staining has been shown to correlate with the severity of the histologic injury. A report of 840 patients with NAFLD found 35% with hepatic iron deposits on staining. A reticuloendothelial system cell pattern of iron staining was linked to advanced disease including portal inflammation, hepatocellular ballooning, steatohepatitis and fibrosis [78]. In addition, increased hepatic iron levels correlate with insulin resistance [79]. Moreover, an improvement in glycemic control correlates with decreases in serum ferritin and hepatic iron concentration [80]. Increased hepatic iron and NASH is linked to a greater prevalence of heterozygosity for the hemochromatosis hepcidin (HFE) mutation in NASH patients [81].

However, not all the evidence supports a link between hepatic iron and advanced NAFLD. HFE homozygosity does not increase the risk for NAFLD [82] and iron accumulation in 65 NASH patients was not related to either overall or liver-related mortality or cirrhosis [83] and there was no improvement in hepatic steatosis or insulin sensitivity following 6 months of phlebotomy [84]. Hormones, as inflammatory markers, are linked to the development of NAFLD. This include adipokines, leptin [85], adiponectin [86], resistin [87] and incretins [88]. Leptin serum levels and liver injury correlated independently of age, BMI and gender. Thus, studies are needed to define whether the hormone plays a major role in the disease status.

## 7. Role of the Intestinal Microbiota

The intestinal microbiota is increasingly being recognized as having an essential role in the maintenance of normal bodily function and being disturbed in disease states. Bacterial products are transported to the liver before any other solid organs via drainage into the portal vein. Thus, it is not surprising that the microbiome has an effect on the development and prognosis of liver disease.

The gut microbiota consists of bacteria, fungi, archaea and viruses [89], although most published data concern bacteria. The gut–liver axis is important in the intestinal immune response, inflammation, both systemic and hepatic, intestinal barrier function and is markedly disrupted in both ALD [90] and NAFLD [91]. The integrity of the intestinal barrier is dependent upon the mucous layer and the commensal microorganisms, secretory IgA and antimicrobial peptides, the epithelial tight junctions and the *lamina propria* with cells of the innate and adaptive immune system [92]. Disruption of the intestinal barrier is central to bacterial translocation form the gut to extraintestinal areas. Passing from the gut to extraintestinal sites occurs upon disruption of the intestinal barrier. 

### 7.1. Alcoholic Liver Disease (ALD)

In alcoholic patients with fatty liver disease or cirrhosis, there is an increase in plasma levels of lipopolysacharides (LPS), which is present in the cell membrane of gram-negative bacteria [93,94]. In human studies, LPS levels are higher in alcoholic cirrhotic patients compared to patients with cirrhosis from other cause [89]. In addition, endotoxemia is higher in patients with more severe cirrhosis as assessed by Child–Pugh score [91]. Thus, it seems that endotoxemia appears early in the course of ALD and is related to the severity of the liver disease. This is due to the disruption of the enterocyte tight junctions by alcohol, which enables bacterial translocation [92,93,94,95,96].

Bacterial translocation also contributes to progression of alcoholic liver disease to cirrhosis and causes severe infections in cirrhotic patients. This may occur by activation of the innate immune system via Toll-like receptors (TLRs). TLRs recognize microbial products termed pathogen-associated molecular patterns (PAMPs). PAMPs include LPS and bacterial peptidoglycan, double-stranded RNA and unmethylated DNA. This complex subject has recently been reviewed [97].

There are also changes in the composition of the intestinal microbiome in ALD. This is due to both intestinal bacterial overgrowth and changes in the composition of the intestinal microbiome [98,99]. This includes decreased levels of anti-inflammatory bacteria such as *Faecalibacterium prausnitzi* and *Bidovacterium* sp. and an increase in *Proteobacteria* sp.

Susceptibility to alcohol-associated liver disease is associated with an imbalance of different bacterial species and a cirrhotic dysbiosis ratio [97,98,99,100]. In addition, there is reduced fungal diversity and *Candida* sp. overgrowth and the production of a pathogenetic endotoxins [101,102]. The bacterial dysbiosis results in specific circulating microbial signatures [99].

Regarding alcoholic liver disease, the relationship between the intestinal microbiome and alcoholic hepatitis has been examined by Puri et al. [103]. They found that the circulating microbiome in heavy drinkers was characterized by a decrease in *Bacteroides* compared to patients with either moderate or severe alcoholic hepatitis. There was, in addition, an increase in *Fusobacteria* in all patients who consumed alcohol. However, there was a higher level of endotoxins in those with severe alcoholic hepatitis. Furthermore, both cirrhotics with high alcohol consumption and those with severe alcoholic hepatitis had activation of the secretion system that has been linked to gram-negative bacterial virulence. This study shows that heavy drinking is an important factor in development of the intestinal dysbiosis and the resulting shift in inferred metabolic function [104,105].

The intestinal microbiome does not, however, explain all the pathogenesis of ALD. A study comparing 18 healthy controls with 48 patients with alcohol use disorder (AUD), 19 with ALD and 29 without, detected was a higher level of serum endotoxins in the AUD group but no difference between the two groups [103]. The distribution of the microbial differences showed major overlaps between the AUD patients with and without ALD. The relative abundance of Bacteroidaceae was lowest in the AUD with ALD group and highest in the healthy control group. Colonic mucosa dysbiosis in the AUD group was not perfectly correlated with ALD. An examination of the fecal metabolome in AUD patients compared with healthy controls found different levels of short chain fatty acids and sulfide as well as a decrease in antioxidant fatty acids, but did not differentiate between AUD and ALD groups [104,105,106].

The dysbiosis related to AUD may be reversible. A study of 60 patients found dysbiosis (reduced Ruminococcaceae abundance and increased intestinal permeability), was only present in 40% of patients [106]. Three weeks after abstinence from alcohol there was an increase in Ruminococcaceae abundance and there was an increase to baseline in the total levels of bacteria. Interestingly, there was an association between increased intestinal permeability and higher levels of anxiety, depression and alcohol craving, which predict future relapse of alcohol abuse. It has been postulated that there is a connection between systemic inflammation to alcohol addiction and mood disorders [106].

### 7.2. Non-Alcoholic Fatty Liver Disease (NAFLD)

The evidence for a causal role of the intestinal microbiome in NAFLD is convincing [107]. Cohousing experiments with mice that have genetic modifications in the inflammasome pathway that predispose to NASH, together with wild-type mice that share their microbiota via coprophagia, results in hepatic steatohepatitis [108]. In addition, fecal microbial transplantation (FMT) from weight-matched obese C57BL/6J mice with or without steatosis to germ-free recipients resulted in the mice receiving stools from steatotic mice developing hyperinsulinemia and steatosis and an increase in hepatic triglyceride content [108]. The steatotic mice had an increase in *Lachnospiraceae bacterium* and *Barnesiella intestinihomin*is [109,110].

The mechanism for the influence of the microbiome on the development of steatosis is similar to that for ALD and has been reviewed by other authors [111]. More recently, studies comparing the composition of the gut microbiota from patients with NAFLD, NASH, cirrhosis and healthy controls have been performed in an attempt to identify gut microbiome signatures and gut-derived metabolites and pathways. [111]. Endogenous alcohol and acetaldehyde production by the microbiome has a role in the pathogenesis of NAFLD. Elshaghabee et al. showed that different microorganisms play specific role in the metabolism of hexoses (fructose and glucose). The researchers show that *Lactobacillus fermentum*, *Weissella confuse* and *Saccharomyces cerevisiae* formed the highest amounts of ethanol when their substrate was glucose. When the substrate was fructose, *S. cerevisiae* and *W. confusa* produced the highest amount of ethanol. Knowing that NAFLD may be caused through ethanol produced from fructose fermentation, it is important to understand the role of intestinal microbiota and microbial metabolism in preventing or causing NAFLD. [112]. Furthermore, these authors showed that oral administration of *W. confusa* resulted in higher fecal and blood ethanol levels and increased liver weight under a high fructose, high fat diet [113].

Elevated levels of alcohol in expired air have also been found in obese female patients with *Candida albicans* overgrowth [114]. Despite having similarities in the pathogenesis, using system biology techniques, researchers have found distinctive molecular signatures for both NAFLD and ALD [115,116].

One of the major drivers of development of the metabolic syndrome is obesity. The interaction between the brain, gut and microbiome is complex and present at all stages of life including modification by antibiotic therapy. This subject has recently been reviewed [117,118,119]. In addition, there appears to be a role for the fungal microbiota in the development of NAFLD [120,121]. Fecal transplantation may have a role in the treatment of the metabolic syndrome in the future [122].

## 8. Endotoxins, Oxidative Status in NAFLD and ALD

As a result of the changes in the intestinal microbiome there is an increase in intestinal permeability in both NAFLD and ALD. This results in an increase in the translocation of bacteria and their products to the liver via the hepatic portal vein. This includes an increase in lipopolysaccharides (LPS) and endotoxins [123]. Both adipokines and endotoxemia are related to the degree of hepatic steatosis in NAFLD [124].

Oxidative stress refers to a redox imbalance secondary to a disbalance between the production and consumption of reactive oxygen species (ROS). NADPH oxidase enzymes (NOX-es) are one of the major sources of ROS. A biomarker for oxidative stress status in healthy individuals is urinary 8-isoprostaglandin F(2α) (8-isoPGF(2α)) [125]. In patients with biopsy-proven NAFLD and the PNPLA38 l148M variant 8-isoPGF(2α) has been shown to be a marker for oxidative stress [126]. There was found to be a correlation between ductular reaction and portal inflammation in the l148M variant carriers compared to non-carriers. In addition, 8-iso- PGF2α. PNPLA3 carriers showed higher steatosis, portal inflammation and hepatic stem cell/progenitor cell niche activation compared to wild-type individuals. Oxidative stress has been shown to be linked to hepatic injury in NAFLD and the progression from steatosis to steatohepatitis, fibrosis, cirrhosis and hepatocellular carcinoma [127]. There is also a link to lipotoxicity [128]. Readers are referred to a recent review of this topic [129].

In ALD, exogenous hepatotoxins may cause an overproduction of reactive oxidative species (ROS), which are generated during microsomal or mitochondrial oxidative stress from incomplete oxygen division and trigger the injury if protective antioxidant capacities are reduced. The organelles targeted are liver mitochondria through lipid peroxidation of membrane structures and the action of free radicals such as superoxide radical, hydrogen peroxide, hydroxyl radical, alkoxyl radical and peroxyl radical. These radicals are covalently binding to macromolecular structural proteins. Liver injury due to chemicals with an organic structure proceeds via the hepatic microsomal cytochrome P450 with its different isoforms.

There is also a role of oxidative stress and inflammatory injury in the pathogenesis of alcoholic liver disease. Chronic alcohol consumption increases NOX4 expression. A NOX-4 inhibitor has been shown to ameliorate liver injury related to alcohol exposure, together with a decrease in the levels of mitochondrial ROS, mitochondrial DNA, respiratory chain complex IV and hepatic ATP. In addition, knockdown of NOX4 decreases mitochondrial membrane potential and decreases mitochondrial superoxide levels, the number of apoptotic cells and lipid accumulation [130].

N-acetylcysteine (NAC) is an antioxidant that is used in the treatment of acetaminophen-induced liver toxicity [131]. It may be a treatment for ALD [132]. In Sprague–Dawley rats receiving alcohol via an intragastric cannula together with an enteral diet, NAC was shown to inhibit lipid peroxidation, decrease the liver injury and maintain the glutathione content [133]. Readers are referred to an extensive review of the role of oxidative stress in the development of ALD [134]. Igomerization domain (NLRs) are groups of receptors [123].

## 9. Clinical Manifestations and Diagnosis of Fatty Liver Disease

### 9.1. Alcoholic Liver Disease

The clinical picture of ALD is varied depending on the severity and chronicity of the alcohol abuse. There is a spectrum of disease ranging from asymptomatic patients to advanced cirrhosis. In addition, there may be acute alcoholic hepatitis. Laboratory biomarkers such as carbohydrate-deficient transferrin (%CDT), gamma—gluthamyl—transferase (gamma-GT), mean corpuscular erythrocyte volume (MCV), EtG (ethylglucuronide) and PEth (phosphatidylethanol) serve as markers for chronic alcohol abuse: in patients with liver disease distinguishing between non-alcoholic and alcoholic origin of the disease [135]. Other laboratory clues to AUD include a macrocytic anemia, elevated ESR and thrombocytopenia.

Accessory investigations are useful but not definitive. The best-known abnormality is an elevation of the AST/ALT ratio. The AST is usually elevated to less than 8 times the upper limit of normal (ULN) and the ALT less than 5 times the ULN. There is no correlation between the amount of elevation and the severity of the ALD [136]. The degree of elevation of AST is typically more than the ALT. This is attributed to a hepatic deficiency of pyridoxal 5′-phosphate, which is a co-factor for ALT activity [136].

The team of Amarapurkar studied 36 NASH individuals, mean age 50.8 years. Fibrosis was present in 30.5% and absent in 69.4% of patients. Using multiple regression and logistic regression analysis, they detected statistical significance for AST, ALT levels and AST/ALT ratio between fibrosis and no fibrosis in 80.6% patients. It is concluded that the AST/ALT ratio may help determine the fibrosis in patients of NASH with diabetes in majority of cases [137].

An AST/ALT ratio above 1 is sometimes present in NASH and is common in NASH cirrhosis. A value greater than 2 is very suggestive of ALD [138,139]. Of 271 patients with biopsy-proven liver disease, 90% of those with an AST/ALT ratio > 2 and 96% of those with a ratio > 2.5 had ALD [140]. In addition, the gamma-gluthamyl transpeptidase (GGT) is often elevated in ALD. A report of 123 patients with AUD revealed that all of the patients with ALD had GGT elevations up to 10 times the ULN. The high level of serum GGT was still found after 8 weeks of abstinence [141].

Imaging studies detect hepatic parenchymal changes and enable a diagnosis of ALD to be made [142] and can be correlated with cytokines and chemokines before and after detoxification [143].

Zhang and his team showed that unenhanced computerized tomography (CT) is capable of detecting and quantifying moderate to severe steatosis. It, however, is inaccurate for diagnosing mild steatosis. The authors also promote the use of dual energy CT for quantifying steatosis. The team also show that magnetic resonance imaging (MRI) proton-density fat fraction is the most precise imaging to quantify liver steatosis [144].

### 9.2. Non Alcoholic Fatty Liver Disease

The majority of the patients with NAFLD are asymptomatic and may not be aware of the diagnosis. The findings on physical examination are similar to that of ALD as described above. There may be a mild to moderate elevation in both ALT and AST, but these tests may also be normal [145]. The AST/ALT ratio is usually less than one. Serum ferritin or transferrin saturation may also be elevated [8,10]. A serum ferritin greater than 1.5 times the ULN is associated with a higher NAFLD activity score and advanced fibrosis [146].

Regarding imaging studies, ultrasound may show hyperechogenicity similar to ALD. A meta-analysis of 49 studies including 4720 patients found a sensitivity of 85% and a specificity of 94% as compared to liver biopsy [147]. However, the sensitivity is decreased in obese patients [148]. CT and MRI do not achieve sufficient sensitivity to detect inflammation or fibrosis [149]. Magnetic resonance (MR) spectroscopy provides a quantitative assessment of hepatic fat.

NAFLD also involves other organ systems as a result of the metabolic syndrome. These include diabetes and cardiovascular disease [150]. In addition there is a connection to diastolic dysfunction [151] and, recently, there has been a report of higher in-hospital mortality in patients with both preserved and reduced ejection fraction heart failure [152].

## 10. Liver Histology in ALD and NAFLD

### 10.1. Alcoholic Liver Disease

Early changes in ALD include macrovesicular steatosis especially in zone 3 [153]. This steatosis may progress to steatohepatitis, with neutrophils also initially in zone 3. Mallory–Denk bodies, which are eosinophilic aggregations within hepatic cytoplasm, are present but may also be seen in NASH [154]. A study from the Veterans Administration found MDB in 76% of those with alcoholic hepatitis and 95% of those with cirrhosis [155]. Fibrosis can eventually develop, again initially in zone 3, and progress to panlobular with chronic alcohol consumption [156]. Eventually, regenerative nodules appear and this stage is thought to be irreversible cirrhosis. Cirrhosis may be either micronodular or macronodular. In some cases, micronodular cirrhosis may progress to macronodular [157].

### 10.2. Non-Alcoholic Fatty Liver Disease

NAFLD is diagnosed when at least 5% of the hepatocytes are steatotic [158]. There may also be iron deposition in the liver [159]. NAFLD may have either steatosis, steatosis with lobular or portal inflammation, or steatosis with ballooning of hepatocytes in the absence of inflammation [160]. The histologic diagnosis of NASH requires hepatic steatosis together with ballooning degeneration of the hepatocytes and lobular inflammation [161]. This is similar to alcoholic steatohepatitis. Mallory–Denk bodies and hepatic siderosis may also be present.

NASH can also exist with other liver diseases including ALD and it is not possible to clearly differentiate between the two [162]. A study of 3581 liver biopsies found steatohepatitis in 5.5% of cases of HCV hepatitis. The authors concluded that the presence of steatohepatitis is a reflection of the frequency of steatohepatitis in the general population [162]. Thus, there is often a similar histological picture of NAFLD or other hepatic diseases, especially ALD. This overlap of ALD and NAFLD with similar clinical and histological characteristics has resulted in a new nomenclature for fatty liver disease.

### 10.3. Revised Nomenclature for Fatty Liver Disease

Since in many countries patients with NAFLD have light or moderate alcohol consumption as well, it has recently been suggested to change the nomenclature of fatty liver disease [163,164]. The new nomenclature emphasizes the metabolic syndrome and suggests there is no safe limit of alcohol consumption for patients with NAFLD [165]. The combination of excessive alcohol consumption (3 drinks or more per day for men and 1.5 for women) in the NHANES III study on USA detected that steatosis increased mortality and the effect was greater in those patients with the metabolic syndrome [166]. Furthermore, low to moderate alcohol consumption is linked to less improvement in both steatosis and steatohepatitis on paired liver biopsies as compared to no alcohol consumption [167]. Furthermore, many of the complications are similar in both ALD and NAFLD. These include an increased risk of liver disease-related hospitalizations [168], and treatment for an increased risk of liver-related mortality [169], increased coronary calcification [170] and increased carotid-intima thickness [171]. There may also be a role of drug hypersensitivity in the development of liver damage by the metabolic syndrome [172,173].

The new terminology for NAFLD is Metabolic Associated Fatty Liver Disease—MAFLD. The term MAFLD has now been accepted by international consensus [174]. The defining criteria of the metabolic risk factors have also been amended and now include a plasma high-sensitivity C reactive protein (CRP) level greater than 2 mg/mL. In this new consensus there is a new diagnosis of dual etiology to include ALD together with MAFLD. 

### 10.4. Treatment of MAFLD

The treatment of MAFLD is mainly by lifestyle changes [175]. This consists of weight loss, a Mediterranean diet, walking 150 min per week, avoiding alcoholic beverages and sweet sugary drinks, and aspirin daily. In addition, bariatric surgery has beneficial effects in terms of reducing obesity and preventing obesity complications [176]. There is no approved pharmacologic treatment of MAFLD despite intense efforts [177]. Although advanced fibrosis is a result of steatohepatitis, the metabolic syndrome causes other organ damage, and cardiovascular complications are common. These can be summarized in a single sentence—weight loss, physical exercise, the Mediterranean diet, and coffee consumption. Weight loss of greater than 10% of body weight has been shown to result in a regression of fibrosis in 81% of patients and weight loss less than 5% causes regression of fibrosis in 45% of patients. Vitamin D deficiency is present in 55% of patients with biopsy-proven NAFLD, and supplementation with vitamin D may be protective against inflammation [178]. Exercise consisting of 150 min of brisk walking per week has a beneficial effect on insulin resistance. A recent review has identified 10 randomized controlled trials that reduce liver fat. The effect is independent of changes in body weight and its effect is greater when combined with weight loss [179].

The high prevalence of MAFLD has generated a large effort by the pharmacological companies to find a successful therapy that is not based just on lifestyle changes. Vitamin E is an antioxidant and was examined for treating NASH in the PIVENS trial, at a dose of 800 IU/day and pioglitazone 30 mg/day. There was a histological improvement but not any significant improvement in fibrosis. It is recommended only for biopsy-proven NASH in non-diabetic patients [180]. An additional therapy is Obeticholic acid. Obeticholic acid is a ligand of Farnesoid X receptor (FXR). It is approved for treatment of primary biliary cholangitis. It has been shown to improve NASH histology without worsening fibrosis [181]. However, it has a deleterious effect on the lipid profile. Since one of the major causes of death for patients with MAFLD is heart disease, this is a major limitation and, in addition, there is a high incidence of pruritus. Non-bile-acid FXR agonists are under investigation [182].

Peroxisome-Proliferator-Activated Receptor (PPARs) Agonists are anti-inflammatory agents that have a role in lipid and glucose metabolism [183]. They are recommended as therapy for biopsy-proven NASH [184,185]. They have side-effects including weight gain and fluid retention. There are ongoing trials with dual PPAR agonist [186,187]. Since insulin resistance is a central factor in developing MAFLD, interest has grown in the use of insulin sensitizers. Metformin has been shown to improve metabolic parameters and decrease serum aminotransferase levels but with no beneficial effect on steatohepatitis [188]. Glucagon-like peptide-1 (GLP-1) agonists have been investigated. Liraglutide resulted in weight loss and improvement in steatohepatitis (and resolution in 29%) but no significant improvement in the fibrosis stage [189]. Semaglutide, a new generation GLP-1agonist, was shown to achieve NASH resolution in a higher percentage compared to controls but again with no change in fibrosis [190,191]. Another treatment for diabetes is sodium/glucose transport protein 2 (SGLT2) inhibitors, which reduce body weight and cause fat loss and have both renal and cardioprotective effects. Dapaglifozin decreases hepatic fat content but does not alter tissue-specific insulin sensitivity [192]. Other treatments being evaluated are probiotics and synbiotics, lipogenesis inhibitors, thyroid hormone receptor beta (TR-beta) agonists, CCL receptor type 2 (CCR2) and type 5 (CCR5 antagonists) and fibroblast growth factor-19 (FGF19) analogue. These are summarized in a recent review [193]. Bone marrow adipocytes, derived from mesenchymal stem cells are a heterogeneous population that interact with hematopoietic precursors and lineage-committed cells. The adipocytes influence commitment and cellular lineage selection by responding to environmental changes, such as obesity, by undergoing hypertrophy and hyperplasia resembling those of peripheral adipocytes. As a result, the bone marrow adipocytes may influence hematopoietic cellular survival, proliferation and preferential differentiation [194].

High-fat diet-induced obesity is associated with a chronic state of low-grade inflammation, which pre-disposes patients to insulin resistance and type 2 diabetes mellitus. Macrophages infiltrate into obese adipose tissue. They release cytokines such as IL-1β, IL-6 and TNFα, creating a pro-inflammatory environment that blocks adipocyte insulin action, contributing to the development of type 2 diabetes mellitus. In lean individuals, macrophages are in an alternatively activated (M2) state. M2 macrophages are involved in wound healing and immunoregulation. Wound-healing macrophages play a major role in tissue repair and homoeostasis, while immunoregulatory macrophages produce IL-10, an anti-inflammatory cytokine, which may protect against inflammation. The functional role of T-cell accumulation has recently been characterized in adipose tissue. Cytotoxic T-cells are effector T-cells and have been implicated in macrophage differentiation, activation and migration. Infiltration of cytotoxic T-cells into obese adipose tissue is thought to precede macrophage accumulation. T-cell-derived cytokines such as interferon γ promote the recruitment and activation of M1 macrophages augmenting adipose tissue inflammation and insulin resistance. Manipulating adipose tissue macrophages/T-cell activity and accumulation in vivo through dietary fat modification may attenuate adipose tissue inflammation, representing a therapeutic target for ameliorating obesity-induced insulin resistance [195,196,197].

### 10.5. Treatment of ALD

The most important component of treatment of ALD is abstinence. Abstinence and high doses of beta-blockers improve long-term rebleeding and mortality in cirrhotic patients after an acute variceal bleeding [198]. A report of 26 patients with hepatic steatosis detected on CT scan who had abstained from alcohol for 6 weeks showed improvement or resolution of hepatic fat infiltration. In addition, two patients had complete resolution after one week of abstinence [199]. Patients who continued to consume excessive alcohol consumption had progression to more advanced disease.

Similarly to NAFLD, there is a role for nutrition in the treatment of ALD. Guidelines include a recommendation to eat multiple meals per day [200]. Liver pathology in ALD is important [201].

In Figure 1 and Figure 2, we present light microscopy and electron microscopy images from a patient with ALD.

We used a special morphometric program Image-Pro Plus (version 5.1 for Windows; Media Cybernetics Inc., Silver Spring, MD, USA). The program permits us to select the lipid droplets in the cells, reports the area of each lipid droplet selected, the diameter of the lipid droplet measured and annotates the image. This enables morphometrical determination of the diameter of the lipid droplets and quantitatively determines the amount of lipids per biopsy. At the bottom of the image there are no more viable cells. These are cells that died by either apoptosis or necrosis. ×60.

There are no approved pharmacological treatments for either ALD [202].

In the following Table 1, we present the common and different risk factors, clinical, pathological and imaging features as well as biomarkers used to identify individuals presenting with alcoholic and nonalcoholic liver steatosis.

## 11. NAFLD, T2D, CVD

Nonalcoholic fatty liver disease, as diagnosed by either liver enzymes or ultrasonography, significantly increases the risk of incident of type 2 diabetes (T2D) and MetS over a median 5 year follow-up. In their metanalysis, Balisteri et al. [203] studied 20 publications (117,020 patients). The included individuals were followed-up for a median of 5 years. They presented an increased risk of T2D. In the same study, 81,411 patients with NAFLD had an increased risk of incident MetS. Studying the Hispanic Latino population with NAFLD, there is an increase of T2D when compared with Caucasians [204].

Targer et al. performed a metanalysis in 16 observational studies including 34,043 adults. In this population, the NAFLD was observed in 36.3%. Over a period of 6.9 years, there were 2600 CVD. The incidence of cardiovascular deaths was >70% for NAFLD patients [205,206]. In a cross-sectional study that enrolled adult 449 TDM subjects from all ethnic groups in Singapore, all subjects underwent hepatic steatosis and fibrosis assessment. Of T2DM patients, 78.72% were diagnosed with NAFLD. From 344 NAFLD individuals, 45 had increased liver stiffness and concomitant hypertension [205].

The T2DM/NAFLD patients represent a higher proportion than the general population. Araujo et. al., discussed the global epidemiology of NAFLD/NASH and the importance of future surveillance [207].

An additional element that is involved in both ALD and MAFLD is vitamin deficiency. Choline is involved in lipid-derived signaling and methylation. There are studies reporting a relation of low choline levels to subclinical organ dysfunction (nonalcoholic fatty liver or muscle damage. In a recent review, Wortmann and Mayr are highlighting the central role of choline within human metabolism [208].

Fatty liver is associated with alcohol abuse, inadequate diet and lifestyle [209]. NAFLD is a result of a combination of several factors such as: the increasing consumption of sugar beverages, processed foods and juices. An important factor is a sedentary lifestyle and environmental toxins. A steatotic liver has a reduced ability to eliminate toxins and hormonal and toxic metabolites. Many exogenous chemicals may have the potential of DILI. Clinical trial protocols should focus on meeting liver injury criteria, exclusion of alternative causes, a robust causality evaluation management, and obtaining liver histology if clinically indicated and of benefit for the patient [210].

NAFLD and the metabolic syndrome are associated with high blood pressure, high blood sugar and significant weight gain. The nutritional triggers of the NAFLD pandemic are not limited to the detrimental effects of a standard diet rich in sugar. Eggs and vegetables (sulfur-rich foods such as garlic, onions) and whole fruit should be recommended. The proportion is four times more vegetables than fruits. Therapeutics should be prescribed wisely and additional over the counter medication should be avoided. In addition, the clinician can ask for laboratory tests confirming metabolic dysfunction, including hemoglobin A1c, fasting glucose, which measures blood sugar; a lipid profile that will examine abnormalities in triglyceride and cholesterol levels; and tissue inflammatory markers (cytokines) [211,212,213,214]. Recent studies show that, under conditions of carbohydrate restriction, fuel sources shift from glucose and fatty acids to fatty acids and ketones, and that ad libitum-fed carbohydrate-restricted diets lead to appetite reduction, weight loss, and improvement in surrogate markers of cardiovascular disease [215].

## 12. Conclusions

Both ALD and NAFLD represent prevalent types of liver disease worldwide. A healthy liver means a healthy metabolism, a healthy cardiovascular system, and an overall healthy body. Fatty liver is the surest prediction of imminent health problems. The liver ability to regenerate is extremely strong. The liver function is of fundamental importance for the body. Detoxification, digestion, hormonal balance, blood sugar regulation and supporting the function of the immune system are only a few of the elements contributing to health.

In a healthy liver, 80% of cells are hepatocytes, with the nonparenchymal cells comprising endothelial cells (8%), stellate (Ito) cells (4%), Kupffer cells (4%), and intrahepatic lymphocytes (4%). Infiltrating T-cells, natural killer (NK)/NK T-cells, and Kupffer cells may contribute to aminotransferase elevations in both ALD and MAFLD. Hepatic NK cells can also contribute to liver injury by a non-antigen-specific TNF-related apoptosis-inducing ligand-mediated pathway. Alcoholic liver disease is directly related to alcohol misuse. Causes include environmental factors and specific genes that affect the risk of alcohol-use disorders, including genes for enzymes that metabolize alcohol, such as alcohol dehydrogenase and aldehyde dehydrogenase; those associated with a low sensitivity to alcohol. On the contrary, causes of MAFLD are related to nutrition and to lifestyle. The effects of microbiome modifications using antibiotics are important factors, while both ALD and MAFLD affect humoral inflammation and progression of advanced liver disease. Microbiota-derived metabolites, such as fatty acids and antioxidants, can be used for the treatment of liver disease. Host-microbial interaction is enabling a new treatment option that may influence metabolic liver disease. Abstinence is the best therapy for ALD.

The clear message is that any alcohol intake, regardless of type of beverage represents a health risk. The nutritional triggers of the NAFLD pandemic are not limited to the detrimental effects of a standard diet. An important element is also a deficiency of choline, which is an essential, vitamin-like nutrient supporting the transport of fat in the body. Its deficiency causes fat to enter the liver, but it cannot leave it. In this review we have presented data regarding ALD, MAFLD and NAFLD, their epidemiology, diagnosis, management, pathophysiology mechanisms, possible future treatments and prevention.

## Figures and Tables

**Figure 1 ijms-23-16226-f001:**
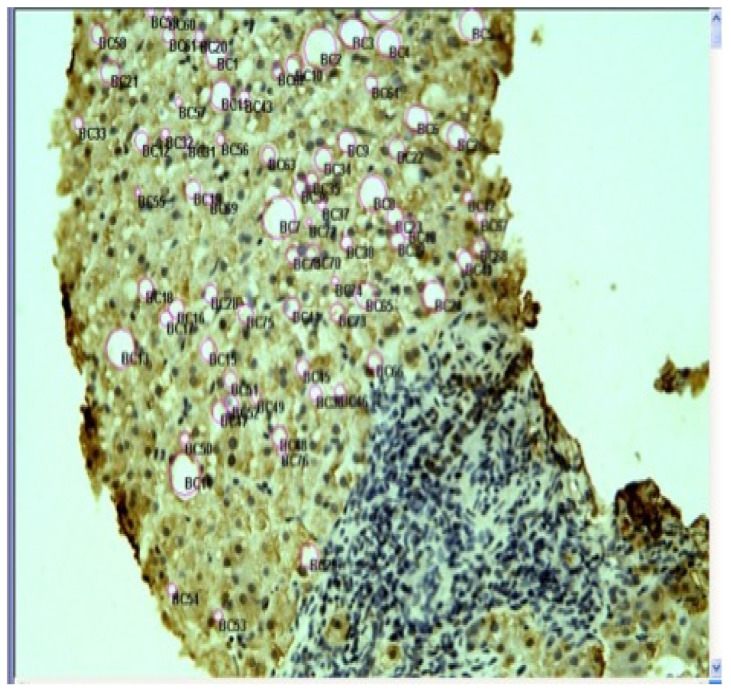
Liver biopsy from a patient with alcoholic liver disease. The biopsy shows hepatocytes with large lipid droplets (balloon cells—BC).

**Figure 2 ijms-23-16226-f002:**
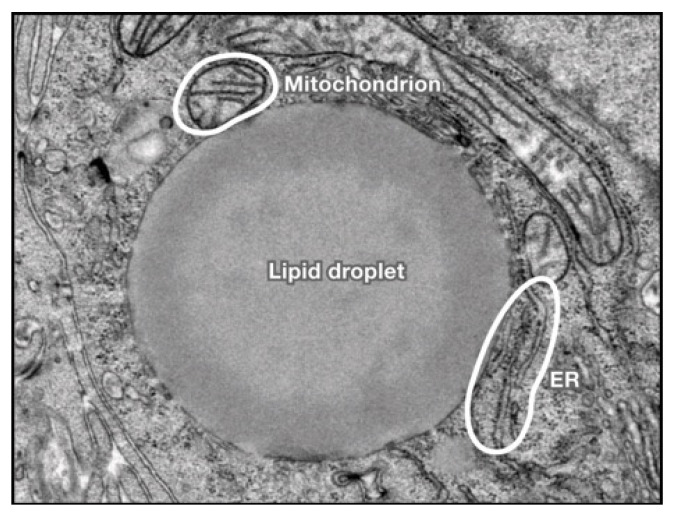
Transmission electron microscopy (TEM) of a hepatocyte from a liver biopsy of an ALD patient. A huge lipid droplet covers the surface of the cell. Endoplasmic reticulum (ER) can be seen. In the upper corner, a mitochondrion with irregular cristae is seen. ×37,500. Unpublished picture adapted from our study- Cameron RG and Neuman MG [157].

**Table 1 ijms-23-16226-t001:** Similarities and differences between NAFLD and ALD.

	NAFLD	ALD
Risk factors	Metabolic syndrome, sedentary lifestyle, processed food, sweet beverages, genetics, intestinal microbiota, type 2 diabetes	Alcohol consumption, genetics, microbiome, endotoxemia
Clinical	Asymptomatic to cirrhosis	Asymptomatic to cirrhosis
Imaging	US/CT- normal to steatosis to cirrhosis MRI?	US/CT- normal to steatosis to cirrhosis; MRI?
Pathology	Steatohepatitis, fibrosis, Mallory–Denk Bodies	Steatohepatitis, fibrosis, Mallory–Denk Bodies
Microbiome	Increase in *Bacteroidetes* sp. and *Proteobacteriae* sp. Decrease in *Firmicutes* sp.	Increase in *Proteobacteriae* sp., decrease in *Faecalibacterium prausnitzi*. Addictive behavior.
Pathogenesis	Inflammatory cytokines, insulin resistance, iron/copper overload, bacterial translocation. carbohydrate metabolism impairment, obesity, sugar excess of dietary long-chain fatty acids	Continuous excess drinking, binge drinking, genetic predisposition, endotoxemia Drugs of misuse combination, malnutrition
Inflammatory cytokines and chemokines	TGF β, TNF α IL-6, IL-8; IL-17; IL-23 RANTES	TGF β, TNF α IL-6, IL-8; IL-17; IL-23 RANTES
Cell death	Apoptosis, necrosis	Apoptosis, necrosis, necroptosis
Treatment	Lifestyle changes, exercise, coffee, dietary changes	Abstinence

## Data Availability

Not applicable.

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
