# Peer review of "Fatty Liver Disease-Alcoholic and Non-Alcoholic: Similar but Different"

_ijms, 2022, doi:10.3390/ijms232416226_

Round 1
Reviewer 1 Report
Authors present a review on the differences between NAFLD and ALD utilizing both clinical perspectives as well as underlying mechanisms. The overall subject areas that the authors touched upon are of importance. However, there are numerous clarifications and elaborations required. I have listed my comments point-by-point below.
Overall, English language editing is needed. Very poor syntax is used throughout making it difficult to read with many awkwardly worded sentences. Also, there are numerous 1-2 sentence paragraphs, I suggest merging these related paragraphs.
Regarding the introduction, utilizing first person writing and unpublished case reports lacks seriousness and is not appropriate for a literature review in the introduction. The same message could be conveyed utilizing published evidence.
Line 145-146: Should replace upgrades to upregulates
Lines 161-163: Expand on these mediators
Line 165: What is the mechanistic role of micro-RNA182, why is it elevated?
Line 203: It is stated “the central abnormality of NAFLD”. I suggest rephrasing this to “underlying cause of NAFLD”. In its current form it is not conveying what is trying to be conveyed.
Lines 206-207: Clarify how insulin resistance is present despite normal glucose tolerance? If referring to specific hepatic insulin resistance, then state so.
Line 209: Standalone paragraph sentence is not needed.
Lines 210-223: This section reads like a list of unconnected surface level facts vs a detailed explanation of the mechanisms. Much greater detail must be provided to tie in these mechanisms with NAFLD.
Lines 225-228: This paragraph is awkwardly placed in this section.
Lines 232-233: Explain the role and functions of these genes and how it relates to NASH.
Lines 234-236: These 2 sentences are not connected. The former sentences states that IR and free radicals should be managed, and the latter sentence discusses TNF, NKs and Caspase 2. It is not clear how these tie in to the former sentence since they are not reactive oxygen species on their own. Much greater elaboration and detail must be provided.
Lines 238-242: These 2 paragraphs are unconnected rom each other and disconnected from the rest of the section. Further, the implications of feroptosis as well as the mediating mechanisms of mentioned hormones is unclear and should be elaborated upon.
Lines 275-276: Can you clarify what these products are?
Lines 310-311: Elaborate on this, what would be considered beneficial signatures?
Lines: 313-314: Do you actually mean "circulating" microbiome?
Lines: 384-385: What about the role of xanthine oxidase and mitochondrial ROS? What activates ROS production?
Line 437: I believe this is an error “with a ratio > 96%”
Lines 455-466: This section again reads like a list of facts vs a cohesive explanation of the clinical manifestation of NAFLD
Line 519 regarding bariatric surgery: Is this strictly to address obesity or MAFLD? Clarification should be provided.
Regarding “Table Summary of the similarities and differences between NAFLD”, label this as Table 1. Then your table title.
Regarding this table, you list "inflammatory cytokines" for the pathogenesis of NAFLD but in ALD write out a list of inflammatory cytokines. It sounds like inflammatory cytokines should simply be stated for both unless there is a unique cytokine profile between NAFLD and ALD.
Regarding section 11 lines 559-582: This section is exceptionally poorly written and does not seem necessary to include. I suggest deleting it.
Regarding the conclusion, it is very surface level. Since the topic is characterizing the differences between ALD and NAFLD, I suggest stating the key differences/similarities between these conditions that can be gleaned from this review.
Author Response
Please find the response in the attachment.

Reviewer 2 Report
The article is of very poor scientific quality and does not introduce anything new that would be of interest to potential readers. First, the authors do not at all explain the meaning of the scientific topic and problem presented. The authors should conclude that a healthy liver means a healthy metabolism, a healthy cardiovascular system, and an overall healthy body. Fatty liver is the surest prediction of imminent health problems. The liver forgives us a lot, and its ability to regenerate is amazing, which the authors do not mention at all. In addition, the liver usually works without publicity or fanfare. It performs its functions, in a hidden way, which, however, are of fundamental importance for health. Detoxification, digestion, hormonal balance, blood sugar regulation and supporting the function of the immune system are some of its nearly 500 tasks. If problems do arise, they are often only diagnosed with routine blood tests for treating other health conditions. The condition in which fat accumulates in the liver and threatens its function is somewhat reminiscent of the problems associated with high blood pressure. People are walking around and they don't know they have a problem. And the authors should also mention it. Until now, fatty liver was most often associated with alcohol abuse. Today we know that a similar disease condition may also be the result of inadequate diet and lifestyle. We are then talking about the so-called non-alcoholic fatty liver disease (NAFLD), which is recently diagnosed on a socially disturbing scale. NAFLD appears to be driven by a combination of several factors (this should be described by the authors). The ever-increasing consumption of sugar and processed foods, but also a sedentary lifestyle and environmental toxins appear to be decisive here. By the way, isn't it weird that liver fat doesn't come from eating fat? The authors should explain this. Even before fatty liver disease occurs, the liver's function changes causing problems throughout the body. Not only is the ability to eliminate toxins and hormonal byproducts diminished, but the immune system and the mechanism that regulates blood sugar are also weakened. Never think of a fatty liver as a separate metabolic dysfunction !!!! After all, NAFLD is closely related to the metabolic syndrome, which in turn is associated with high blood pressure, high blood sugar and significant weight gain. It is also a precursor to diabetes, stroke and cognitive decline. The nutritional triggers of the NAFLD pandemic are not limited to the detrimental effects of a standard diet. An important element is also a deficiency of choline, which is an essential, vitamin-like nutrient supporting the transport of fat in the body. Its deficiency causes fat to enter the liver, but it cannot get out of it. Also, without enough choline, we overproduce fat and don't eliminate it !!! The drain is clogged, and authors should remember that the best sources of choline are whole eggs and the liver. Another factor influencing fatty liver is a sedentary lifestyle. Exercise has countless health benefits, but it protects the liver quite simply; they just burn more fat than when you do nothing. Finally, the liver is also the main organ detoxifying our body, and we have known for a long time how much toxins we are currently exposed to. Scientists are still trying to understand the interactions that threaten us as a result. Meanwhile, the accumulation of toxins and the need to neutralize them is a serious and daily challenge for our liver. And it is not miraculous cleansing diets, after which we return to our old tastes as absolved, that are the solution to this situation. Less radical but permanent changes will be better here: Giving up processed foods by the food industry whenever possible. Paying attention to what we drink. Each colored drink bodes bad for the liver. Clean water is always the best solution. Best to eat vegetables and fruit whole. We should eat four times more vegetables than fruit. Let's not squeeze any juices out of them, our liver doesn't like it very much. Do not eat over-the-counter drugs and ask your doctor if prescription drugs are necessary. Processing drugs is a lot of effort for the liver, and we should limit it to what is really necessary. The list could be longer, of course, there is a group of cruciferous vegetables especially welcomed by the liver, there are herbs, such as the ubiquitous dandelion in our fields, and sulfur-rich foods such as garlic, onions and eggs. And this is what the authors should definitely mention. The excess fat around the viscera is certainly also deposited in the liver. Maybe this is a good time to ask your doctor for a blood test and a so-called liver tests. In addition, we can also do tests confirming metabolic dysfunction, which usually go hand in hand with fatty liver (NAFLD). These tests include hemoglobin A1c, which is an indicator of blood sugar levels over the last three months; fasting glucose, which measures blood sugar levels after a fasting period of at least eight hours; a lipid profile that will examine abnormalities in triglyceride and cholesterol levels; and a C-reactive protein that tracks tissue inflammation. The authors should also describe the risk factors, causes of the disease and how it should be treated in terms of all the latest scientific developments.
Author Response

(The authors gave the same response as above.)

Round 2
Reviewer 1 Report
Authors have addressed most comments. A few additional minor comments:
A citation is needed for the first few sentences of the introduction since evidence is referenced.
The conclusion needs additional refinement, it is not written well and does not summarize the differences between the 2 states very well.
Author Response
Thank you very much for the review.
We took your advise and make the requested changes.